# The canonical E2Fs together with RETINOBLASTOMA-RELATED are required to establish quiescence during plant development

Magdolna Gombos[1,8], Cécile Raynaud[2,3,8], Yuji Nomoto[4], Eszter Molnár[1], Rim Brik-Chaouche[2,3], Hirotomo Takatsuka[4], Ahmad Zaki[5], Dóra Bernula[1], David Latrasse[2,3], Keito Mineta[4], Fruzsina Nagy[1,6], Xiaoning He[2,3], Hidekazu Iwakawa [4], Erika Őszi[1], Jing An [2,3], Takamasa Suzuki[7], Csaba Papdi[5], Clara Bergis[2,3], Moussa Benhamed [2,3], László Bögre[5], Masaki Ito[4] & Zoltán Magyar [1✉]

Maintaining stable and transient quiescence in differentiated and stem cells, respectively, requires repression of the cell cycle. The plant RETINOBLASTOMA-RELATED (RBR) has been implicated in stem cell maintenance, presumably by forming repressor complexes with E2F transcription factors. Surprisingly we find that mutations in all three canonical E2Fs do not hinder the cell cycle, but similarly to *RBR* silencing, result in hyperplasia. Contrary to the growth arrest that occurs when exit from proliferation to differentiation is inhibited upon RBR silencing, the *e2fabc* mutant develops enlarged organs with supernumerary stem and differentiated cells as quiescence is compromised. While E2F, RBR and the M-phase regulatory MYB3Rs are part of the DREAM repressor complexes, and recruited to overlapping groups of targets, they regulate distinct sets of genes. Only the loss of E2Fs but not the MYB3Rs interferes with quiescence, which might be due to the ability of E2Fs to control both G1-S and some key G2-M targets. We conclude that collectively the three canonical E2Fs in complex with RBR have central roles in establishing cellular quiescence during organ development, leading to enhanced plant growth.

[1] Institute of Plant Biology, Biological Research Centre, H-6726 Szeged, Hungary. [2] Université Paris-Saclay, CNRS, INRAE, Université Evry, Institute of Plant Sciences Paris-Saclay (IPS2), 91190 Gif sur Yvette, France. [3] Université de Paris Cité, CNRS, INRAE, Institute of Plant Sciences Paris-Saclay (IPS2), 91190 Gif sur Yvette, France. [4] School of Biological Science and Technology, College of Science and Engineering, Kanazawa University, Kakuma-machi, Kanazawa 920-1192, Japan. [5] Royal Holloway, University of London, Department of Biological Sciences, Egham, Surrey TW20 0EX, UK. [6] Doctoral School in Biology, Faculty of Science and Informatics, University of Szeged, H-6726 Szeged, Hungary. [7] College of Bioscience and Biotechnology, Chubu University, Kasugai, Aichi 487-8501, Japan. [8] These authors contributed equally: Magdolna Gombos, Cécile Raynaud. ✉email: magyarz@brc.hu

In multicellular organisms, both the entry into and the exit from cell division cycle is developmentally regulated. There is a tight balance among cells proliferating in meristematic regions of emerging young organs, stem cells that are in a reversible quiescent status with a capacity to re-entry into cell division upon stimulation, and the most common cellular state of terminally differentiated cells formed after permanent exit from the cell cycle[1–3]. The principal regulation of cell proliferation by highly conserved molecular mechanisms is well established[4], but how the cell cycle machinery is controlled in quiescent cells to maintain transient and stable repression of mitosis remains unclear.

Evidence from animal model organisms indicates that the RETINOBLASTOMA (Rb)-E2F transcriptional regulatory pathway plays a central role in coordinating proliferation and quiescence[2,5]. Binding of Rb to the E2F-dimerisation partner (DP) heterodimers can convert these complexes to transcriptional repressors, while phosphorylation of Rb by Cdk-cyclin complexes releases E2Fs and their ability to activate transcription. Accordingly, animal E2Fs can function as bistable switches on their target genes, determining whether the cells enter into the cell cycle or exit to establish quiescence[6].

The Rb-E2F pathway is remarkably conserved in plants, and it is well established that the plant RETINOBLASTOMA-RELATED (RBR) coordinates cell proliferation and differentiation by regulating the activity of three canonical E2F transcription factors (E2FA, E2FB and E2FC) that form dimers with one of the two DP proteins (DPA and DPB)[7–10]. The three non-canonical E2Fs (E2FD, E and F) function independently of DP and RBR[11,12]. Canonical E2Fs are structurally related to their animal counterparts, and consensus E2F-binding sites were identified in many cell cycle regulatory genes[11,13]. Their C-termini contain the transactivation domain overlapping with the RBR-binding region[14,15] and thus function as transcriptional activators or repressors depending on whether they are in complex with RBR[16,17]. Overexpression studies suggest that E2FA and E2FB function as activators, but recent findings showed that they could also repress genes specifically involved in embryo and seed development[18]. Surprisingly, genetic studies showed that E2F mutations either in single, double or even triple mutants did not universally compromise cell proliferation[17–20].

It has been established that loss of function *rbr* mutants are gametophytic lethal due to abnormalities during meiosis and gametophyte development[21–23]. Manipulation of RBR level during sporophyte development severely affect the balance between proliferation and differentiation; ectopic overexpression of RBR led to diminishment of root and shoot meristems, while the consequence of RBR silencing is the inhibition of differentiation and increased number of stem cells[7,24]. Conditional RBR silencing strongly represses leaf growth by stimulating the excessive proliferation of stem cells in the stomatal lineage and inhibiting terminal differentiation[8,25], and triggers cell death[7,26]. It was suggested that upon RBR silencing the E2Fs are de-repressed, which is responsible for the above described developmental defects. However, the recent description of a fully viable, although partially sterile mutant plant, in which all three canonical E2Fs carry a T-DNA insertion (a triple *e2fabc* mutant) questions whether E2Fs are indeed essential for the activation of cell proliferation[20,27]. This is surprising because there is ample evidence that plant E2Fs are the primary effectors of RBR just like in animals[28,29].

To understand the role of E2Fs for the developmental regulation of cell cycle, in this work we analysed the *e2fabc* triple mutant in detail for cell proliferation defects and for the deregulation of E2F targets. Surprisingly, just as upon RBR silencing, we observed hyperplasia in the *e2fabc* triple mutant, suggesting that E2Fs are not required for maintaining cell proliferation, but rather have a repressive role. RBR silencing leads to growth arrest during seedling establishment due to the inhibition of this developmental transition and of cellular differentiation[9]. This is not the case for the *e2fabc* mutant, which shows hyperplasia during embryonic and post-embryonic organ development with delayed cell cycle exit and unstable quiescence both in stem and in terminally differentiated cells. The production of excess number of cells in a developmental setting naturally does not mean that all cells uncontrollably proliferate, as these cells become incorporated into an enlarged organ. Some of these extra cells are the result of continued proliferation, e.g., more rounds of amplifying divisions of stomata meristemoids. In other cases, we observed differentiated pavement cells to form cell division planes. It is equally possible that these differentiated pavements cells retain their ability to proliferate or they re-enter cell division after they attained quiescence. We also established that the three E2Fs and RBR have largely overlapping gene targets and cell cycle genes were markedly upregulated in *e2fabc* mutant seedlings. We conclude that the three canonical E2Fs are collectively involved in cell cycle exit and in establishing transient and stable cellular quiescence by forming repressor complexes with RBR.

## Results

**The triple *e2fabc* mutant displays hyperplasia, leading to enlarged organs and plant stature.** Several alleles of T-DNA insertion lines have been described for canonical E2Fs, all of which disrupt the carboxyl terminus containing the entire transactivation and RBR-binding domains and they, except *e2fa-2* also lost the MARKED BOX (MB) region, that mediates the interactions between E2Fs and various other transcription factors in animals (Supplementary Fig. 1a)[18,30]. By using specific antibodies in immunoblot assays, we confirmed the loss of the corresponding E2F proteins as well as the accumulation of a truncated E2FA protein in the *e2fab* double and in the *e2fabc* triple mutants (Supplementary Fig. 1b, c). These E2F mutations, however, did not affect DPA, DPB or RBR protein level in the triple *e2fabc* mutant seedlings (Supplementary Fig. 1d). As described previously, the triple *e2fabc* mutant is partially sterile, mostly due to abnormal gametophyte development (Fig. 1a), but the reason for this is not yet known[27]. In addition, at a very low frequency, there is a so far unnoticed anomaly of supernumerary nuclei in the *e2fabc* mutant embryo sacs (2%; *n* = 240), a characteristic phenotype for *rbr* mutant ovules (Fig. 1a)[21,31]. Further similarities to *rbr* mutant phenotypes are the twin embryo sacs[31], that may originate from two megagametophyte mother cells (Fig. 1a)[27]. These excessive proliferation phenotypes suggest that the canonical E2Fs may also have common repressor functions with RBR at some stages of gametophytic development.

To our surprise, the embryo and plant that came through the viable gametes of *e2fabc* triple mutant was overall normal but showed an increased plant size at the adult stage (Fig. 1b, c) as well as longer roots (Fig. 1d, e) and larger plantlet and rosette sizes (Supplementary Fig. 2a–e). Importantly, another triple *e2fabc* mutant obtained by crossing the *e2fab* double with the *e2fc-2* T-DNA insertion mutant (Supplementary Fig. 1a), gave identical phenotypes with the original *e2fabc* mutant line (Supplementary Fig. 2f–h).

Due to the above-described sterility, the *e2fabc* mutant only produced few seeds, but these were significantly larger and heavier than WT (Fig. 1f)[20], containing enlarged embryos (Fig. 1g and Supplementary Fig. 3) that consisted of smaller and more numerous cells than WT and double *e2fab* mutant (Fig. 1h–j and Supplementary Fig. 3a–e)[17]. Contrary to the triple *e2fabc* mutant embryo, double *e2f* mutant embryos (*e2fab*, *e2fbc* or *e2fac*) did

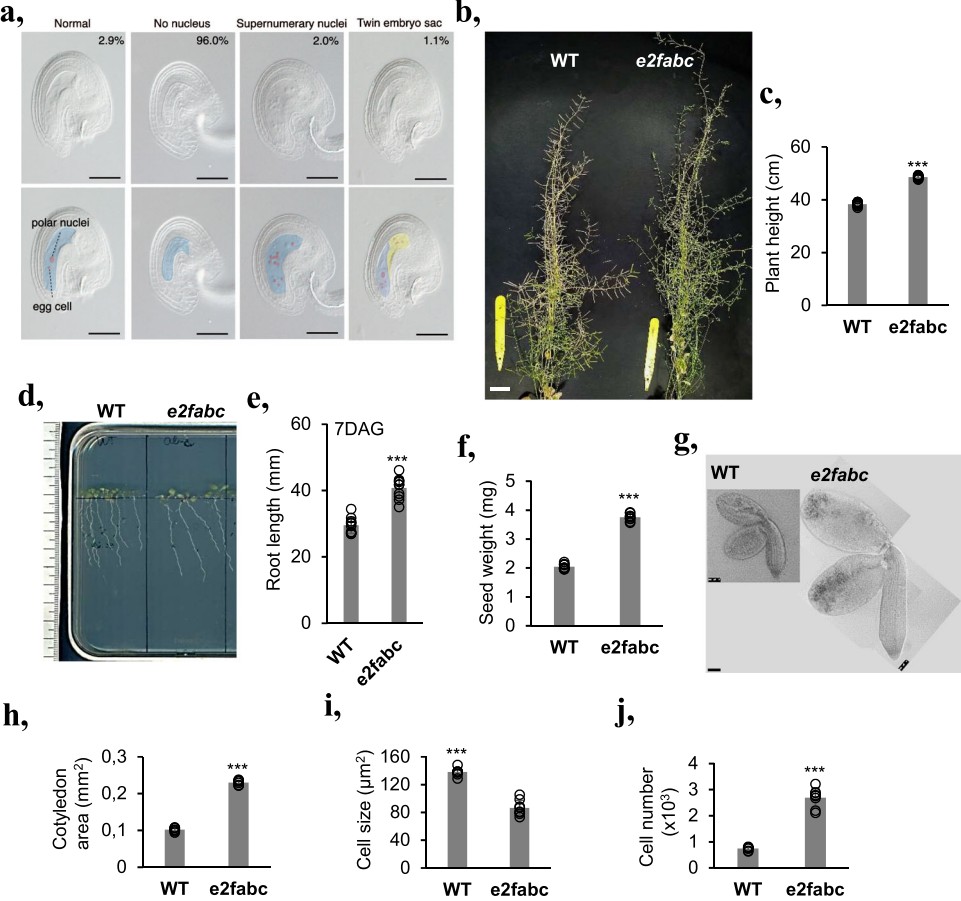

**Fig. 1 The *e2fabc* mutant displays hyperplasia. a** Gametophytic defects observed in *e2fabc* triple mutants (highlighted in the lower panel of pictures). DIC images of cleared ovules showing embryo sacs with non-distinguishable nuclei (96% of images), a minority of normal-looking embryo sacs that account for the few viable seeds produced by the mutant (2.9%), but also embryo sacs with supernumerary nuclei (2,0%) or even twin embryo sacs (1,1%). The latter two are excessive proliferation phenotypes that were also reported in *rbr* mutants. Scale bars = 50 μm. **b** A representative picture of mature plants showing that the *e2fabc* mutant has shorter siliques and is taller than the WT. Pictures were taken 30 DAG. Bar is 1,5 cm. **c** Graph showing average height of WT and *e2fabc* plants ($N = 10$ samples in each). ***$P \leq 0.001$ indicates statistically relevant differences between the mutant and the WT (two-tailed paired *t* test between the mutant and the WT). Root length is increased in *e2fabc* mutants as shown by representative pictures of plantlets at 7 DAG (**d**), and quantification of root length (**e**), data are average +/− standard deviation ($n = 3$ biological replicates; $N = 10$ samples in each), ***$P \leq 0.001$ indicates statistically relevant differences between the mutant and the WT (two-tailed paired *t* test between the mutant and the WT). **f** Seed weight of *e2fabc* mutant is increased (weight of seeds is per 100 seeds, $n = 10$/line). ***$P \leq 0.001$ was considered significant between the WT and the mutant (two-tailed paired *t* test between the mutant and the WT). **g–j** *e2fabc* embryos are larger than those of WT and consist of more and smaller cells. **g** representative pictures of dissected embryos from mature seeds (Bar: 200 μm), (h) cotyledon area, (i) epidermal cell size, (j) epidermal cell number in embryonic cotyledon. Cell size was calculated using ImageJ. Sample size $N \geq 200$ cells/image. For all graphs, data are average +/− standard deviation ($n = 3$ biological replicates; $N = 8$ sample in each) and ***$P \leq 0.001$ (two-tailed paired *t* test between the mutant and the WT) indicate statistically relevant differences.

not show hyperplasia (Supplementary Fig. 3e, f). Likewise, developing leaves of the *e2fabc* mutant seedlings became gradually larger than those of WT, until they reached 1.5 times the size of WT leaves at 20 DAG (Fig. 2a, b). Cellular analysis revealed that cell size was comparable with WT at early leaf development but failed to reach WT values at 16 DAG (Fig. 2c). Consequently, epidermal cell number was significantly higher in *e2fabc* mutants than in the WT (Fig. 2d). In addition, we observed some fully differentiated lobbed pavement cells with straight division planes in the *e2fabc* mutant, but not in WT, indicating their re-entry into cell division or continue proliferation while simultaneously differentiating (Fig. 2e, f and Supplementary Fig. 4a). Introducing a nuclear targeted CYCB1;2-YFP, that marks mitotic cells until telophase, into the triple *e2fabc* mutant further supported that enlarged, lobbed pavement cells divide, while in the control WT, CYCB1;2-YFP signal is never present in these type of cells (Supplementary Fig. 4b). Similar divisions of

differentiated leaf epidermal cells was reported when regulators of this pathway like CYCD3;1 or E2FB was ectopically expressed in tobacco leaves[32,33]. The number of these extra divisions continuously rose and became the highest in the most advanced leaves at 16 DAG, resulting in smaller sized cells than in the WT (Fig. 2d–f), further suggesting that differentiated cells undergo extra cell divisions. As a result, the *e2fabc* mutant leaves contained almost twice as many cells (48%) as the control WT at 16 DAG (Fig. 2d). In some large pavement cells multiple cell division planes could be detected, indicating repeated cell division cycles (Supplementary Fig. 4a). Leaf cells dispersed among lobbed pavement cells, round and smaller than 60 μm² are considered to be stomatal meristemoids[34]. The number of these cells was significantly increased in the expanding *e2fabc* mutant leaf at 12 and 16 DAG (Fig. 2e, g), suggesting that also the cell cycle exit is compromised in stomata stem cells in the *e2fabc* mutant. Consistent with the excessive proliferation the expression levels

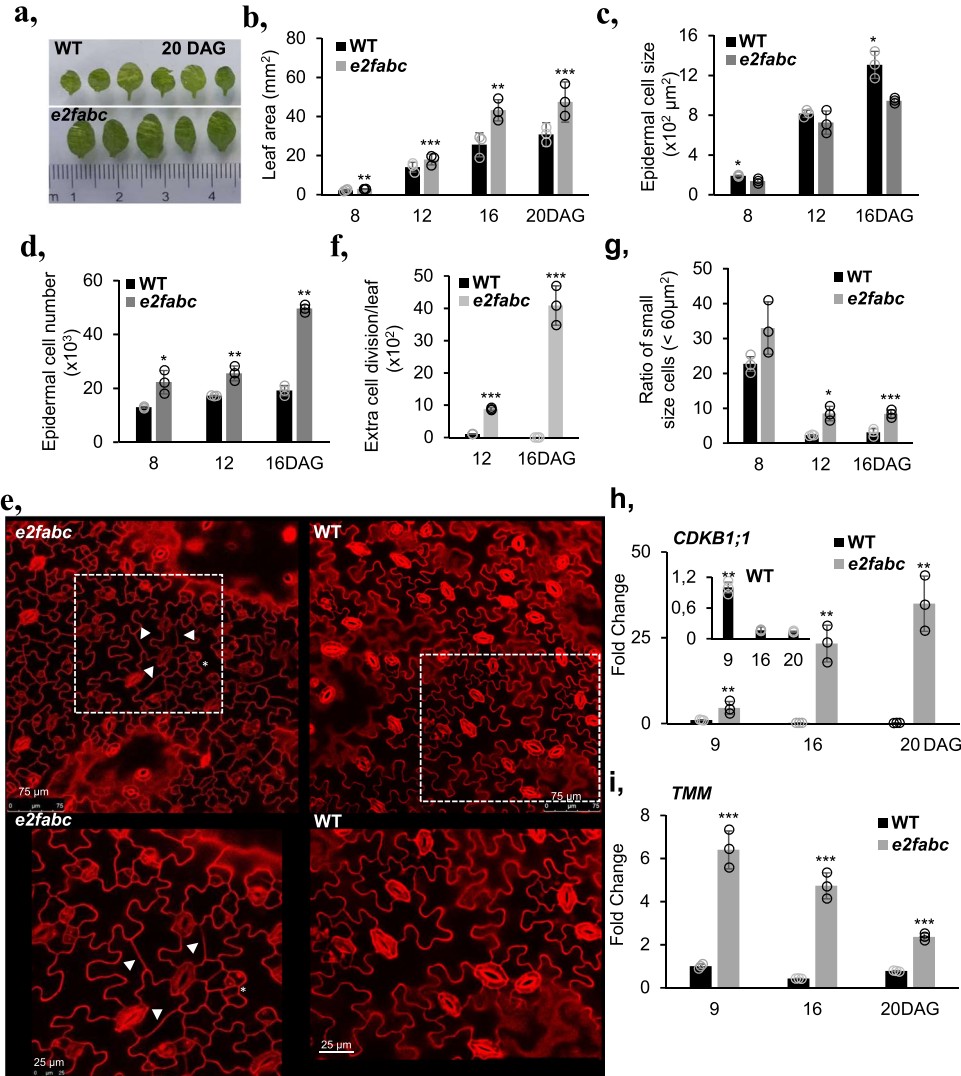

**Fig. 2 Hyperplasia in developing leaves of the *e2fabc* mutant. a–d** First leaves of *e2fabc* mutant are enlarged due to prolonged proliferation, such as in stomata meristemoids and due to possible re-entry into cell division, such as in differentiated puzzle shaped pavement cells. **a** representative pictures of the first leaves of WT plants or *e2fabc* mutants at 20 DAG. **b** quantification of leaf area, (**c**) quantification of cell area, and (**d**) quantification of cell number in the leaf epidermis (first leaf pair). For all graphs, data are average +/− standard deviation *n* = 3 biological replicates, *N* = 10 samples in each. **P ≤ 0.001, ***P ≤ 0.0001 indicates statistically relevant differences between the mutant and the WT (two-tailed paired *t*-test between the mutant and the WT). **e–g** The increased cell number is derived from prolonged proliferation of stomata meristemoids and re-entry into cell division in differentiated puzzle shaped pavement cells in the *e2fabc* mutant. **e** Confocal microscope image of leaf epidermis of WT and *e2fabc* plants stained with propidium iodide (PI). Clustered stomata meristemoids (*) and division of differentiated puzzle-shaped cells (arrowheads) are observed in the *e2fabc* mutant. White dashed boxes outline the epidermal region of leaf in WT and *e2fabc* and magnified below. (Bar:75 μm and 25 μm). **f** Quantification of extra cell division events (*n* = 3 biological replicates, *N* = 6 samples in each), (**g**) proportion of small cells corresponding to stomata meristemoids. Data are mean +/− sd., *n* = 3 biological replicates, *N* = 5 samples in each and sample size ≥600 cells *P < 0.05, ***P < 0.001 (two-tailed, paired *t* test between the mutant and the WT at a given developmental time point). **h, i** The cell cycle gene *CDKB1;1* and the stomata development gene *TMM* are elevated in *e2fabc* mutant compared to the wild-type. Expression of the two genes was monitored by qRT-PCR. Inset shows *CDKB1;1* expression in the WT. Values represent fold changes normalised to the value of the relevant transcript of the wild type at 9DAG, which was set arbitrarily at 1. Data are means +/− sd., *n* = 3 biological repeats. **P < 0.01; ***P < 0.001 (two-tailed, paired *t* test between the WT and the mutant at a given time point). Abbreviations and primer sequences are listed in Supplementary Table 1.

of the S-phase-related *MCM3* (Supplementary Fig. 4c) and the mitotic *CDKB1;1* (Fig. 2h), continuously increased in the *e2fabc* mutant during leaf development whereas these were strongly reduced or only detectable at the youngest stage of 9 DAG in the WT. To see whether the elevated expression of S-phase genes is related to endocycle or overproliferation in the triple *e2fabc* mutant compared to WT, we measured the nuclear DNA content of the first leaf pairs at early, when normally there is no endocycle, and at a later developmental stages, when control WT

leaves enter endocycle (8 and 14 DAG, respectively), as well as in cotyledons, which is known to have high degree of endoreduplication at 10DAG (Supplementary Fig. 4d). As expected, in WT the 8C DNA content started to appear in true leaves at 14 DAG and reached 16C in the cotyledons, while in the triple mutant the 4C DNA level was the highest both in 14 DAG leaf and in the cotyledon, indicating a G2 arrest and block of endocycle. Similar effect on the leaf nuclear DNA content was reported when RBR level was silenced, further supporting that RBR functions together

with E2Fs[7]. The elevated expression of two genes associated with the stem cell lineage of stomatal meristemoids: *TOO MANY MOUTH* (*TMM*; Fig. 2i), and *SPEECHLESS* (*SPCH*; Supplementary Fig. 4c) was also in agreement with the observed phenotype. Moreover, both RBR protein amount and its phosphorylation increased and remained at high levels in the *e2fabc* mutant leaf during its development in comparison to WT, supporting that quiescence was not stable in the triple *e2fabc* (Supplementary Fig. 4e). In contrast, the leaf size of *e2fab* double mutant was comparable to WT (Supplementary Fig. 5a, b). Additionally, the expression pattern of the mitotic *CYCB1;1* in the developing *e2fab* double mutant leaf was similar to WT, while it was significantly upregulated in the triple *e2fabc* mutant (Supplementary Fig. 5c). Taken together, these data suggest that the three canonical E2Fs together are essential to allow cell cycle arrest to establish both stable and transient quiescence.

**Canonical E2Fs maintain quiescence through the recruitment of RBR to repress cell cycle-related genes.** To further understand the underlying mechanism for the repressor role of E2Fs, we investigated global gene expression changes in the triple *e2fabc* mutant seedlings (Supplementary Data 1). 1766 genes were significantly upregulated and only 647 were downregulated in *e2fabc* compared to the WT (adj *P* value < 0.01). Additionally, Gene Ontology (GO) terms associated with upregulated, but not with downregulated genes, were highly enriched with functions linked to cell cycle control (Fig. 3a). To go further, we analysed the expression changes of two gene categories: the G1/S transition specific genes consisting of the canonical E2F targets, that were defined as genes misregulated in *E2FA* overexpressing lines, and harbouring a consensus E2F binding motif in their promoter[13], and the G2/M-specific genes, respectively[35]. Among the genes upregulated in *e2fabc*, we found significant enrichment for canonical E2F target genes (*P*-value < 2.2e-16), but interestingly G2/M-specific genes were also affected (*P*-value = 0.0001401, Fig. 3b). On the other hand, neither categories were enriched among the downregulated genes in *e2fabc* mutant (Fig. 3b, *p* = 0.3336 and 1 respectively). Similarly, the scatter plot comparison of transcript levels in *e2fabc* and wild type seedlings showed general upregulation of the canonical E2F target genes (Fig. 3c).

Strong upregulation of cell cycle genes is also observed when RBR function is inhibited[8,9]. To test whether E2FA, B, C and RBR actually target the same loci, we performed chromatin immunoprecipitation (ChIP) followed by sequencing (ChIP-seq) using plants expressing translational GFP fusions under the control of the native promoters[16,17,36,37]. Binding of all four proteins occurs mostly just upstream from transcriptional start sites of genes (Supplementary Fig. 6a), and targets identified here for E2FA and RBR largely overlapped with previous reports (Supplementary Fig. 6b, c)[38], validating the reliability of our data. Pearson correlation analysis (Fig. 3d), as well as plots representing 2 by 2 comparisons of distances separating defined peaks from the closest TSS (Fig. 3e) revealed that the four proteins have very similar binding profiles genome-wide, as further illustrated by a representative chromatin enrichment profile in Fig. 3f.

The ability of canonical E2Fs and RBR to target the same set of genes, together with the observation that the simultaneous loss of E2FA, B and C triggers excessive proliferation like RBR silencing, suggest that one key function of E2F proteins during vegetative development could be the recruitment of RBR to impose cell proliferation arrest. We therefore monitored the presence of RBR-DPB complexes in the *e2fabc* mutant, compared to *e2fab* mutants in which E2FC is still intact and its heterodimeric E2FC-DPB

form is able to form complexes with RBR. As expected, RBR was in complex with DPB in the *e2fab* double mutant (Fig. 4a), but strongly reduced in the *e2fabc* mutant lacking E2FC, supporting our hypothesis that RBR recruitment to its targets is affected in these mutants. The residual RBR binding in the *e2fabc* mutant might be mediated by the MB region retained on E2FA in the case of *e2fa-2* mutant allele.

Because the strong RBR co-suppression line (*RBRcs*) is arrested during seedling growth[9] (Supplementary Fig. 7a), we analysed the cotyledons for excessive proliferation and compared them to the *e2fabc* mutant (Fig. 4b, Supplementary Fig. 7). As in developing leaves, puzzle-formed pavement cells and guard cells in their regular bicellular forms could form in the cotyledon of *e2fabc* mutants, but meristemoids appeared clustered and differentiated pavement cells showed large number of straight cross walls, while neither of these ectopic proliferations were present in the WT (Fig. 4b, Supplementary Figs. 7b and 4a, Supplementary 8c, d). In the *e2fabc* cotyledon epidermis, straight cross walls could be detected in nearly all pavement cells supporting that quiescence was neither efficiently established nor maintained in these otherwise committed epidermal cells (Fig. 4b and Supplementary Fig. 8d). By contrast, most of the epidermal cells in *RBRcs* were small embryonic-like (Fig. 4b; Supplementary Fig. 7c, d), and many dead cells could be visualised (Fig. 4b; Supplementary Fig. 7c, d)[8,9]. Cell size was markedly shifted from large cells towards smaller cells, and cell number was increased in both *RBRcs* and *e2fabc*, although much more in *RBRcs* (Supplementary Fig. 8a–c). Consistently, the expression of key cell cycle regulators, DNA repair genes and embryonic genes was markedly increased in *RBRcs* line, and to a lesser extent in *e2fabc* mutant (Supplementary Fig. 9a). Extra cell divisions could also be visualised both in cotyledons and leaves of *e2fabc* triple mutant through the accumulation of the CYCB1;2 protein labelled with YFP (Supplementary Figs. 4b and 10). By contrast, photosynthesis-related genes that fail to be activated in *RBRcs* line due to defects in cell differentiation, were expressed at comparable levels in *e2fabc* and WT plants (Supplementary Fig. 9a). These findings match the observation that cell differentiation is delayed but not compromised in *e2fabc* mutant whereas the *RBRcs* line is severely growth arrested because cell differentiation is prevented[9]. Interestingly, late embryonic *LEC2* and *ABI3* genes were expressed at a comparable high level in both *RBRcs* and *e2fabc* mutants, and additionally seed storage 12S globulin was accumulated in the *e2fabc* seedlings suggesting that embryonic genes are repressed by RBR after germination probably in complex with E2Fs (Supplementary Fig. 9b).

We also compared cell proliferation defects in the root meristem of *e2fabc* mutants and in a weaker RBR loss of function line, where RBR levels are reduced by an artificial microRNA (*amiRBR*)[39]. Pulse labelling with 5-ethynyl 20-deoxyuridine (EdU) revealed a similar increase of S-phase cell number in the *e2fabc* mutant and the *amiRBR* line in comparison to the WT (Fig. 4c, d). Furthermore, we observed supernumerary undifferentiated columella stem cells in the *e2fabc* root cap, as well as supernumerary cortex/endodermis initials (Fig. 4e, f and Supplementary Fig. 11a), as previously shown in an RBR silenced line[7]. Mitosis, as followed by the YFP-tagged CYCB1;2 marker was clearly present in small dividing epidermal cells and the supernumerary columella initials, providing further evidence for their sustained ability to divide (Supplementary Fig. 10–11b). The lack of signal in quiescent centre (QC) compared to the surrounding cells when using prolonged EdU labelling allows to visualise the low frequency of S-phase in cells in QC, as shown in WT, while both in the *e2fabc* triple mutant and in *amiRBR* the QC cells were positive for EdU (Fig. 4g). Quiescence of stem cells was shown to protect them against DNA damage[39]. As it was

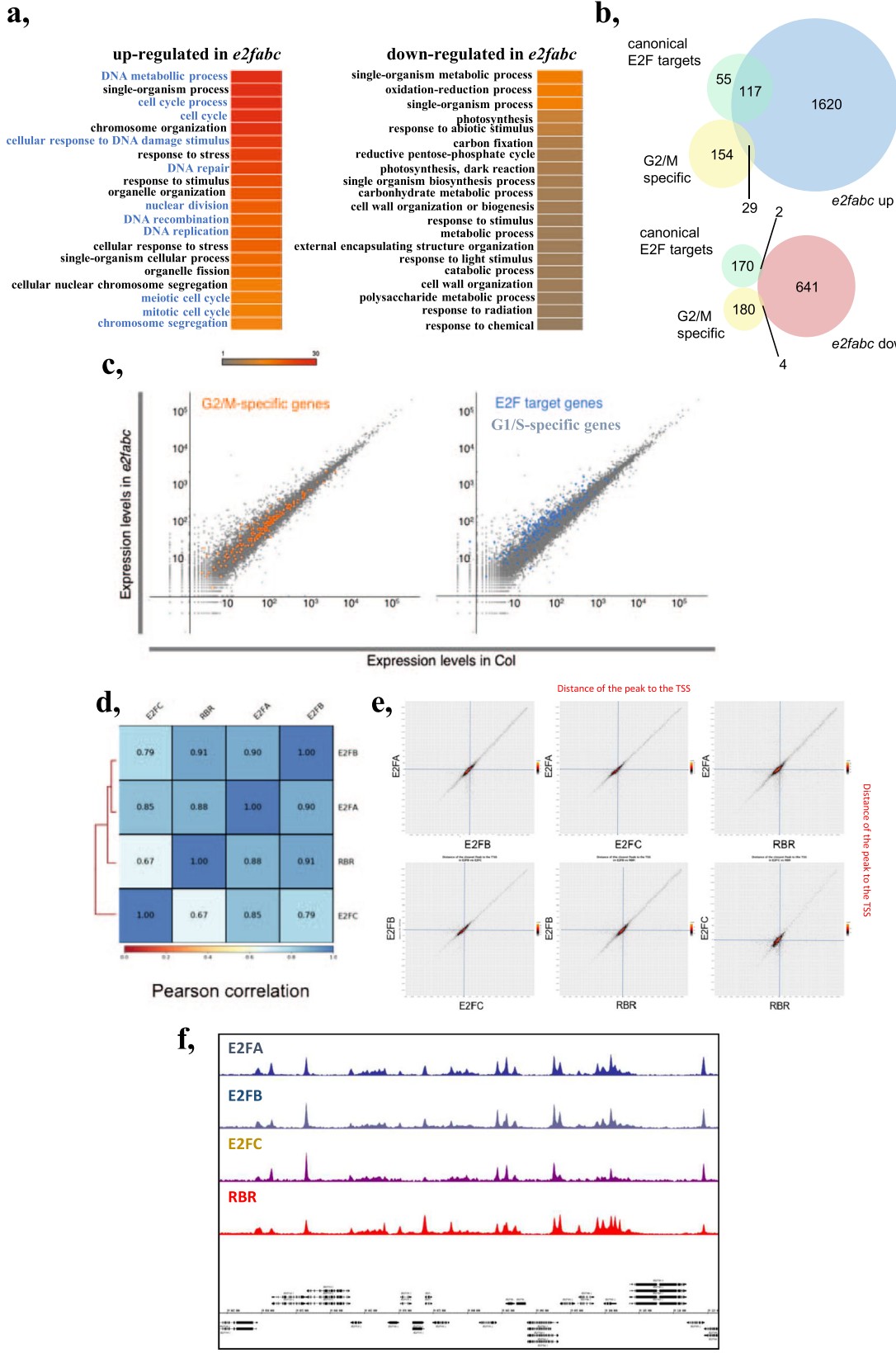

shown before for *amiRBR*, we found an increased sensitivity of root stem cells of the *e2fabc* line compared to WT to the DNA damaging agent cisplatin (Supplementary Fig. 12c). The RBR silenced lines also display spontaneous cell death within the root meristem, but this is not the case for the *e2fabc* line (Supplementary Fig. 12a, b). Together, these data indicate that proliferation activities of the root meristem and specifically of the stem cells increased in the triple *e2fabc* mutant root, as is the case in *amiRBR* line, providing evidence for the role of E2F/RBR complexes in the maintenance of the quiescent state.

**Fig. 3 E2FA, B and C together act as repressors of cell cycle genes. a–c** Cell cycle genes are globally upregulated in *e2fabc* triple mutant. **a** GO term enrichment found amongst genes up- or downregulated in *e2fabc* triple mutant. The colour scale corresponds to the -log$_{10}$ of the FDR. **b** Venn diagram showing the overlap between up- or downregulated genes in *e2fabc* mutant and G1/S or G2/M cell cycle regulators. **c** Scatterplot highlighting the expression of G2/M specific (orange) and canonical E2F target (blue) genes in *e2fabc* mutant compared to the WT. **d–f** E2FA, B, C and RBR have largely overlapping target sites genome-wide. **d** Pearson correlation of binding sites identified by ChIP-seq for E2FA, B, C and RBR. **e** plots showing 2 by 2 comparison of the position of binding sites for E2FA, B, C and RBR on their targets. Plots are centred on the TSS of target genes. **f** Example ChIP-seq profiles to illustrating the similarities among E2FA, B, C and RBR.

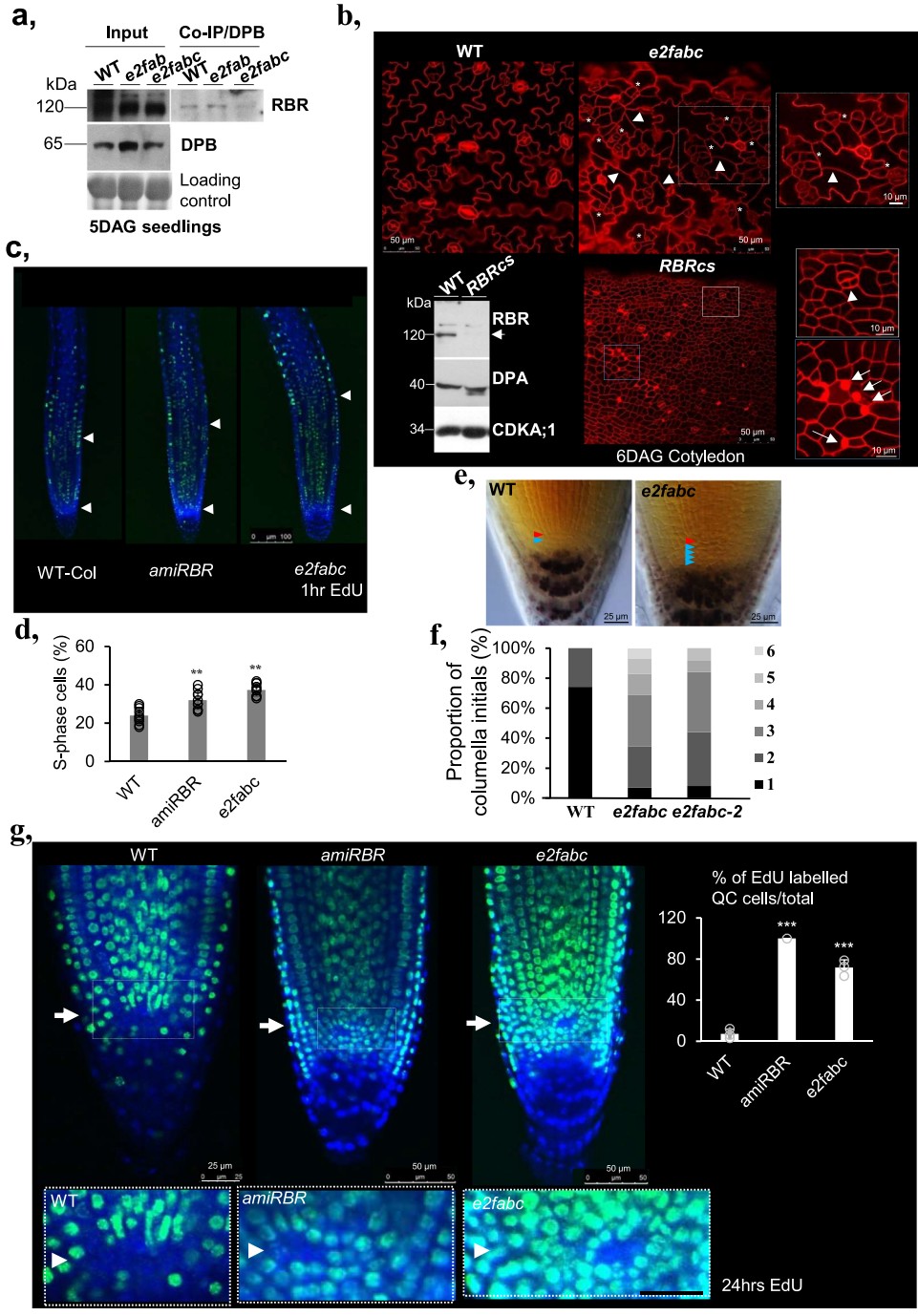

**E2Fs and MYB3Rs participate in common multiprotein DREAM-complexes but have unique regulatory roles in the cell cycle**. MYB3Rs together with E2FB, E2FC, and RBR, but not E2FA have been implicated in the repression of cell cycle genes as part of large multiprotein complexes, known as DREAM[26,37,40]. We therefore investigated the interplay among E2Fs, RBR and repressive MYB3Rs in relation to their target gene regulation. We defined target genes as those detected with the highest confidence (*q*-value < 0.01 enrichment > 3; Supplementary Data 2). As expected, lists of genes bound by each E2F

**Fig. 4 E2F/RBR complexes cooperate to control cellular quiescence. a** RBR-DPB association is strongly reduced in the *e2fabc* mutant. Protein complexes were immunoprecipitated from protein extracts obtained from WT, *e2fab* or *e2fabc* mutant seedlings grown at 5 DAG using anti-DPB antibody. Comparable levels of both RBR and DPB can be detected in all genotypes (left, Input). However, although RBR could be co-IP-ed with DPB using protein extracts of WT or *e2fab* seedlings, amount of RBR immune-precipitated from *e2fabc* protein extracts were dramatically reduced (see also in Supplementary Fig. 14). **b** Comparison of the cellular phenotypes observed in the cotyledon epidermis of *e2fabc* triple mutant and *RBRcs* line. Asterisks indicate clustered stomata meristemoids, arrowheads point to extra cell division events in puzzle-shaped differentiated pavement cells found only in *e2fabc*. In *RBRcs* line all epidermal cells are small embryonic-like. White and blue boxes outline the epidermal region of cotyledon in *e2fabc* and *RBRcs* magnified on the right side. Arrowhead in *RBRcs* shows a dividing guard cell, while white arrows mark dead epidermal cells. Western blot in the left corner shows the lack of RBR in the *RBRcs* in comparison to the WT seedlings at 6DAG, while DPA and CDKA;1 amounts were comparable between them. Arrowhead shows RBR protein, and molecular weight markers are indicated on the left side (see also in Supplementary Fig. 14). **c, d** *e2fabc* and *amiRBR* lines display enlarged root meristems. Roots of the indicated genotypes were labelled with EdU for 60 min. **c** EdU staining of root tips of WT, *e2fabc* and *amiRBR* lines. White arrowheads mark the position of QC and the end of the meristematic zone indicated by enlarged nuclear size in the root epidermal cells. Bar: 100 μm. **d** Percentage of EdU-positive S-phase cells relative to DAPI-stained nuclei in the root meristems of WT, *amiRBR* and *e2fabc* lines. Average of 10 roots were imaged for each genotype. Data are average +/− standard deviation ($n = 3$ biological replicates, $N = 10$ samples in each). **$P < 0.01$ (two-tailed, paired *t* test between the mutant and the WT). **e, f** *e2fabc* triple mutant with supernumerary columella initials. **e** microscopy images of lugol-stained root tips of WT and *e2fabc* mutants (Bar: 25 μm). Differentiated root cap cells can be identified by amyloplast accumulation. The red arrowhead points to the QC and blue arrowheads indicate layers of columella initials. **f**: quantification of the proportion of roots with 1, 2, 3, 4, 5 or 6 layers of columella initials in WT, *e2fabc* or *e2fabc-2* mutants. **g** Cell proliferation is reactivated in the QC of *e2fabc* and *amiRBR* lines. Plants were incubated in EdU for 24 h. Representative images are shown on the top panel and the white arrows point at the QC (Bar: 25 and 50 μm). White boxes outline the QC region of root meristems and magnified in below (Bar: 20 μm). White arrowheads mark the QC. The graph below shows the proportion of roots showing EdU labelling in the QC. Average of 35 roots were imaged for each genotype. Data are average +/− standard deviation ($n = 3$ biological replicates, $N = 35$ samples in each). ***$P < 0.001$ (two-tailed, paired *t* test between each mutant and the WT).

were largely overlapping, irrespective of the combination of isoforms looked at (Supplementary Fig. 13a). To collectively deal with target genes of E2FA, E2FB and E2FC, we defined a gene category representing general E2F targets, that is characterised by being amongst the most significant targets of at least two E2Fs (hereafter called E2F(2)). E2F(2)-, RBR- and MYB3R3-bound genes showed dramatic overlap in any combinations (Supplementary Fig. 13b), defining 7 gene categories depending on the combination of factors that bind to them (Supplementary Data 3). *EMR* (bound by E2F, MYB3R3, and RBR) corresponded to 637 genes (Fig. 5a). For most of these genes, peaks of ChIP signals exactly coincide for E2FA, E2FB, E2FC, RBR and MYB3R3 at the site of transcriptional initiation (Supplementary Fig. 13c). GO terms associated with *EMR* genes were highly and exclusively enriched with categories related to cell cycle (Fig. 5a). Enrichment of cell cycle-related GO terms was observed both for *EMR* and *EM*, and less prominently in *M*, but not at all in other categories (Fig. 5a). Both the G1/S E2F targets and the G2/M-specific genes comprise a significant fraction of *EMR* and *EM* genes (Fig. 5b). This suggests that genes with cell cycle-related functions are preferentially regulated under the dual control of E2F and MYB3R transcription factors.

We then analysed the effects of E2F and MYB3R mutations on the expression of genes in each category as defined by our ChIP-seq data. For this purpose, RNA-seq gene expression datasets of *e2fabc* and *myb3r135* triple mutant were utilised. Genes within the *EMR* and *EM* groups were frequently upregulated in either *e2fabc* or *myb3r135* mutants, but very rarely in both (Fig. 5c). Therefore, we concluded that in spite of the fact that MYB3Rs and E2Fs can be present in the same protein complex, and they are recruited onto the same promoter site, on a particular target gene only one or the other predominantly regulates the expression. This unique mode of regulation can be also confirmed for canonical E2F target and G2/M-specific genes, which are preferentially influenced by E2Fs and MYB3Rs, respectively, but much less frequently by both factors (Fig. 5d). Interestingly, the propension of genes to be influenced either by E2Fs or MYB3Rs correlated with the presence of consensus motifs for E2F or MYB3R binding, respectively (Fig. 5e, f). Altogether, these data indicate that E2F and MYB3R transcription factors could bind to common cell cycle genes, but they

specifically regulate the expressions of genes holding their characteristic E2F or MSA binding sequences, respectively.

While MYB3Rs are specifically targeted and regulate mitotic genes containing MSA elements, E2Fs can regulate cell-cycle genes involved both in G1-S and G2-M transitions (Fig. 3b), an example is CDKB1:1 (Fig. 2h), a critical controller of G2-M and the switch from mitosis to endoreduplication[41]. Interestingly, the activating MYB3R, MYB3R4 is also a direct target of canonical E2Fs and is upregulated in the *e2fabc* mutant leaves at all analysed time points, whereas expression of the repressor MYB3R3 remained unchanged (Fig. 5g). Accordingly, the MYB3R4 target, *KNOLLE* with a role in cytokinesis, was also upregulated (Fig. 5g). These suggest that E2Fs may also regulate mitosis indirectly through the activating MYB3Rs. The ability of RBR and the three canonical E2Fs to regulate both G1/S and G2/M transitions may explain why only upon their loss the quiescence is broken, but not upon MYB3Rs that only act on mitotic genes.

## Discussion

The E2F-Rb regulatory pathway is a pivotal conserved regulator of cell proliferation both in animal and plant cells, acting at the critical transition point of G1 to S phase, known as restriction point. The textbook model is that E2Fs, when Rb-bound, restrict cell proliferation by preventing the expression of key cell cycle genes. This repression is relieved through Rb phosphorylation by CDKs, the activity of which is regulated by a plethora of signalling inputs. Therefore, it is surprising that genetic knockout of all three canonical E2Fs in Arabidopsis can produce viable plants with a functional cell cycle. One possibility could be that the distantly related DP and E2F-like DELs (DEL1-3 or E2FD-F) might replace canonical E2Fs to activate cell cycle genes, but they considered to be repressors without the ability to induce gene expression[42]. Here we show that while none of the canonical E2Fs are fully required for activation of cell cycle genes, but collectively they are responsible for their repression and thereby to establish both transient quiescence in stem cells and stable quiescence in differentiated cells leading to supernumerary cell numbers in all organs examined. This scenario resembles to what has been established for the mammalian activator E2Fs (E2F1-3), as on the one hand they were also found dispensable for cell proliferation

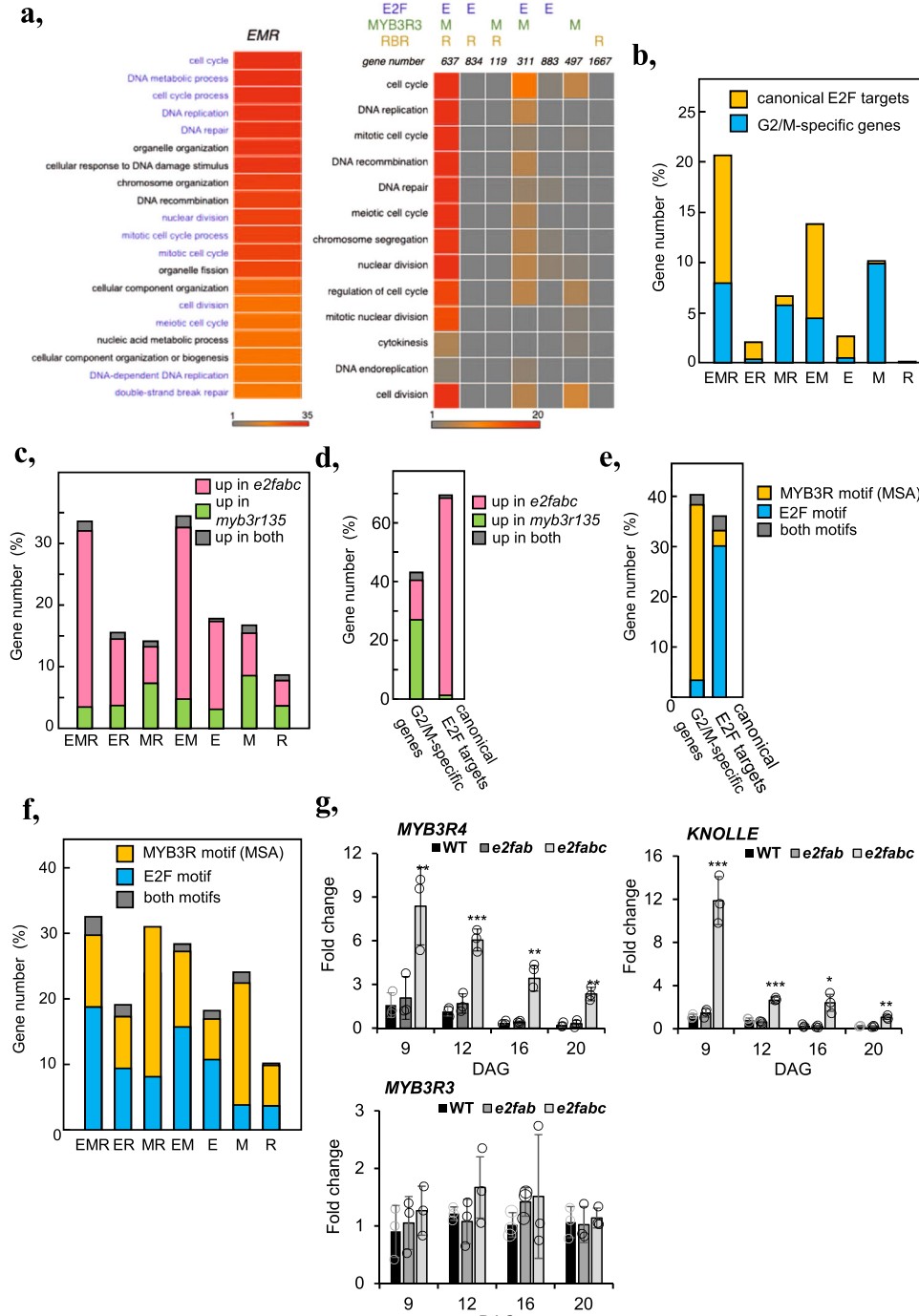

**Fig. 5 Targets of E2Fs and repressive MYB3Rs largely overlap but they control distinct sets of cell cycle genes. a** GO term enrichment analysis shows that genes bound by E2Fs, RBR, and MYB3Rs are significantly overrepresented for cell-cycle related genes, whereas this enrichment is not as obvious for genes targeted by 2 or only one of these factors. **b** G1/S and G2/M-related genes are highly enriched in the common targets of E2Fs, MYB3Rs and RBR (EMR). This graph represents the percentage of G1/S (yellow), or G2/M (blue) genes amongst the six categories of genes defined according to their binding by E2F (E), MYB3Rs (M) RBR (R). **c** Genes targeted by E2Fs, RBR and/or MYB3R are preferentially regulated either by E2Fs or by MYB3Rs. This graph represents the percentage of genes in each category that are upregulated either in *e2fabc* or *myb3R1,3,5* mutants, or upregulated in both. **d–f** Cell-cycle related genes targeted by E2Fs, RBR and/or MYB3R are preferentially regulated either by E2Fs or by MYB3Rs. **d** graph representing the percentage of G1/S or G2/M genes that are upregulated either in *e2fabc* or *myb3r1,3,5* mutants, or upregulated in both. **e** Graph representing the percentage of G2/M and G1/S genes harbouring canonical E2F- or MYB3R-binding motifs in their promoters. **f** Graph representing the percentage of genes harbouring a canonical E2F or MYB3R binding motif in their promoters within the 6 categories defined according to the binding profiles of E2Fs, MYB3Rs and RBR. **g** The activator MYB3R4 but not the repressor MYB3R3 is upregulated in *e2fabc* mutant. Expression of MYB3R4, MYB3R3 and the MYB3R4 target and G2/M marker KNOLLE was monitored by qRT-PCR in the first leaf pairs of WT, *e2fab* and *e2fabc* mutants at 9, 12, 16 and 20 DAG. Values represent fold changes normalised to the value of the relevant transcript of the WT at 9DAG, which was set arbitrarily at 1. Data are means $+/-$ sd., $n = 3$ biological repeats. **$P < 0.01$, ***$P < 0.001$ (two-tailed, paired $t$ test between the WT and the mutant at a given time point). Abbreviations and primer sequences are listed in Supplementary Table 1.

while on the other hand they form complexes with Rb in differentiating cells to repress cell cycle genes[43].

Much of the previous studies have focused on the functional differences among the three canonical E2Fs[16,44–47]. E2FA and E2FB are portrayed as transcriptional activators while E2FC as a repressor. Moreover, it appears that they work through different mechanisms, only E2FB and E2FC but not E2FA are part of DREAM-like multi-subunit protein complexes[37,40]. Cell proliferation in single *e2fa*, *e2fb* or *e2fc* mutants is largely normal and these plants only display mild phenotypes, e.g., the compromised meristem maintenance and reduced formation of lateral root primordia in *e2fa* mutant, and a slight increase in cell number during leaf development in *e2fb* mutant and an *e2fc* silenced line[17,18,47,48], suggesting that the distinction between activator and repressor classes of E2Fs is more nuanced. Interestingly, full loss of function *e2fab* double knock-out line could not be obtained[49], but this appears to be allele specific, as combining the *e2fb* mutant with *e2fa-2* is viable and did not show proliferation defects either during embryo or leaf development[18,19]. The hyperplasia in the *e2fabc* triple was not characteristic for any of the *e2fc* double mutant combinations with *e2fa-2* (*e2fc-1/e2fa-2*) or *e2fb-1* (*e2fc-1/e2fb-1*), suggesting that it is not E2FC, but the action of the three E2Fs together that is responsible for the repression. This was also the case when combining the *e2fa/e2fb* double mutant with two different *e2fc* knockout lines, both displayed hyperplasia, mostly due to the inability of maintaining quiescence. This suggests that the three canonical E2Fs are collectively required for the repression of cell cycle genes even if they act through different molecular mechanisms[17,18]. The principal component of repressor complexes on all canonical E2Fs is RBR and it was demonstrated that all three canonical Arabidopsis E2Fs strongly interact with it[16,17,37]. However, we have seen a stunning difference between the phenotypes of RBR silencing and the triple *e2fabc* mutant, both showing hyperplasia during organ development, but only upon RBR silencing and not in the *e2fabc* mutant the cell differentiation is repressed. A similar inhibition of differentiation occurs when CYCD3;1, the regulatory component of the major RBR kinase is overexpressed[50]. Transient over-expression of CYCD3;1 and CYCD4;2 in nonproliferating tobacco epidermal leaf cells was shown to induce the re-entry into cell division, demonstrating that quiescent leaf cells maintain their competence to divide[32]. These suggest that the de-repression of active E2Fs upon RBR silencing or phosphorylation interferes with differentiation and maintenance of quiescence. Because through the dimerisation partner, DPB, we still could detect a tiny amount of RBR association in the triple *e2fabc* mutant, presumably through the conserved MB region that was retained on E2FA in the *e2fa-2* allele, we cannot rule out that a small residual E2F-RBR complex is able to repress cell proliferation to an extent that lead to cell cycle exit and differentiation[51]. However, the RBR association in the *e2fabc* is far less than in the double *e2fab* mutant, supporting that the main RBR binding activity of E2Fs is at the C-terminal RBR-binding domain, which in the case of *e2fab*, is presented on E2FC[16].

On the one hand, RBR repression on E2Fs is the pivotal regulator of the restriction point in G1 and in response to developmental cues can determine the G1 length and whether cells can exit to differentiation. In this scenario tuning E2F activation of cell cycle genes by RBR is the critical mechanism. On the other hand, mutation of all three E2Fs impairs both the RBR-E2F corepressor function and the transcriptional activation of cell cycle genes. The comparison of RBR silencing and *e2fabc* mutant lines indicates that E2F activator function is important to maintain proliferation through preventing cell cycle exit and the onset of differentiation, while maintaining quiescence is primarily regulated by the RBR-E2F corepression. However, quiescence was not disrupted in each of the studied organs and cell types the same way in the *e2fabc* mutant, reflecting that different cell types have various levels of quiescence dependent on the developmental context, developmental regulation, and positional organisation of stem cells and cell files, as it was demonstrated in animals[3]. Leaves and roots for example have different temporal and spatial proliferation patterns under the control of the E2F-RBR regulation[6,16] suggesting that this pathway is involved to establish and maintain quiescence in both organs but whether alone or in combination with other regulatory pathway is not known yet. In addition, our results show that plant E2Fs, like their animal counterparts, are the primary effectors of RBR in nearly every process where RBR is involved (Fig. 6a). The transient stem cell quiescence is most sensitively abrogated in the stomata lineage when RBR is silenced or the canonical E2Fs are mutated, as indicated by the abundance of small cells characterised as stomata stem cells and by the continuous upregulation of stomata stem cell factors, *TMM* and *SPCH*[8]. Interestingly, RBR, in an E2F-independent manner, by direct interaction with the stomata terminal differentiation transcription factor, FAMA, was also shown to regulate the late steps of differentiation within the stomata cell linage[52]. However, in the *e2fabc* mutant we have not seen the characteristic phenotype of the *fama-1* mutant, called fama-tumours, clusters of small, narrow epidermal cells that are unable to go through the final stages of stomata differentiation. RBR and E2Fs can also directly regulate genes involved in late embryonic differentiation programs, e.g., *LEC2* and *ABI3*, which are direct E2F and RBR targets and are upregulated both when RBR is silenced or in the *e2fabc* mutant, but the developmental transition of seedling establishment is blocked only in the former[9]. Therefore, it is likely that as during leaf emergence, the developmental arrest in seedlings, when RBR is silenced, is due to the deregulated cell cycle.

It was shown both in animals and plants that quiescence protects long-lived stem cells against stress and toxicities[39,53,54]. In agreement, we found that stem cells are sensitive to genotoxic stress both upon RBR silencing and in the *e2fabc* mutant. RBR and E2Fs were also implicated to directly regulate the expression of DNA damage genes and cell death. In this respect we observed a much more pronounced spontaneous cell death with RBR silenced cells than in the *e2fabc* mutant, which correlated with the level of induction of DNA damage genes, suggesting that both co-repression and E2F transactivation function contributes to this regulation.

RBR, E2FB and E2FC together with the mitotic MYB3Rs can be part of large repressor complexes, called DREAM[37]. Here we show that RBR, E2Fs and MYB3Rs are also recruited to largely overlapping sets of cell cycle target genes, but still maintain a specific regulatory function on E2F targets or mitotic genes dependent on whether the MSA or E2F target sequence is present in these promoters. We have shown previously that mutation of the three repressor MYB3R (*myb3r1/3/5*) in Arabidopsis resulted in excess cells in developing organs but exclusively within the meristems[40]. Interestingly, mitotic genes like *CYCB1;2* were found to be ectopically expressed also in differentiated leaf cells, but this did not result in mitotic reactivation, possibly because E2F-RBR complexes are still present, and capable of repressing genes involved in the G1/S transition, which may explain why quiescence is efficiently maintained in the myb3r1/3/5 mutant (Fig. 6b). In contrast, in the *e2fabc* mutant, where the E2F target genes are de-repressed, we find that both the transient quiescence in stem cells and stable quiescence in differentiated cells are broken. Although E2Fs primarily regulate genes with G1-S function, they were also shown to control the expression of key G2-M regulatory genes, like the plant specific CDKBs, supporting that E2Fs regulate both cell cycle transitions[45]. Moreover, E2Fs

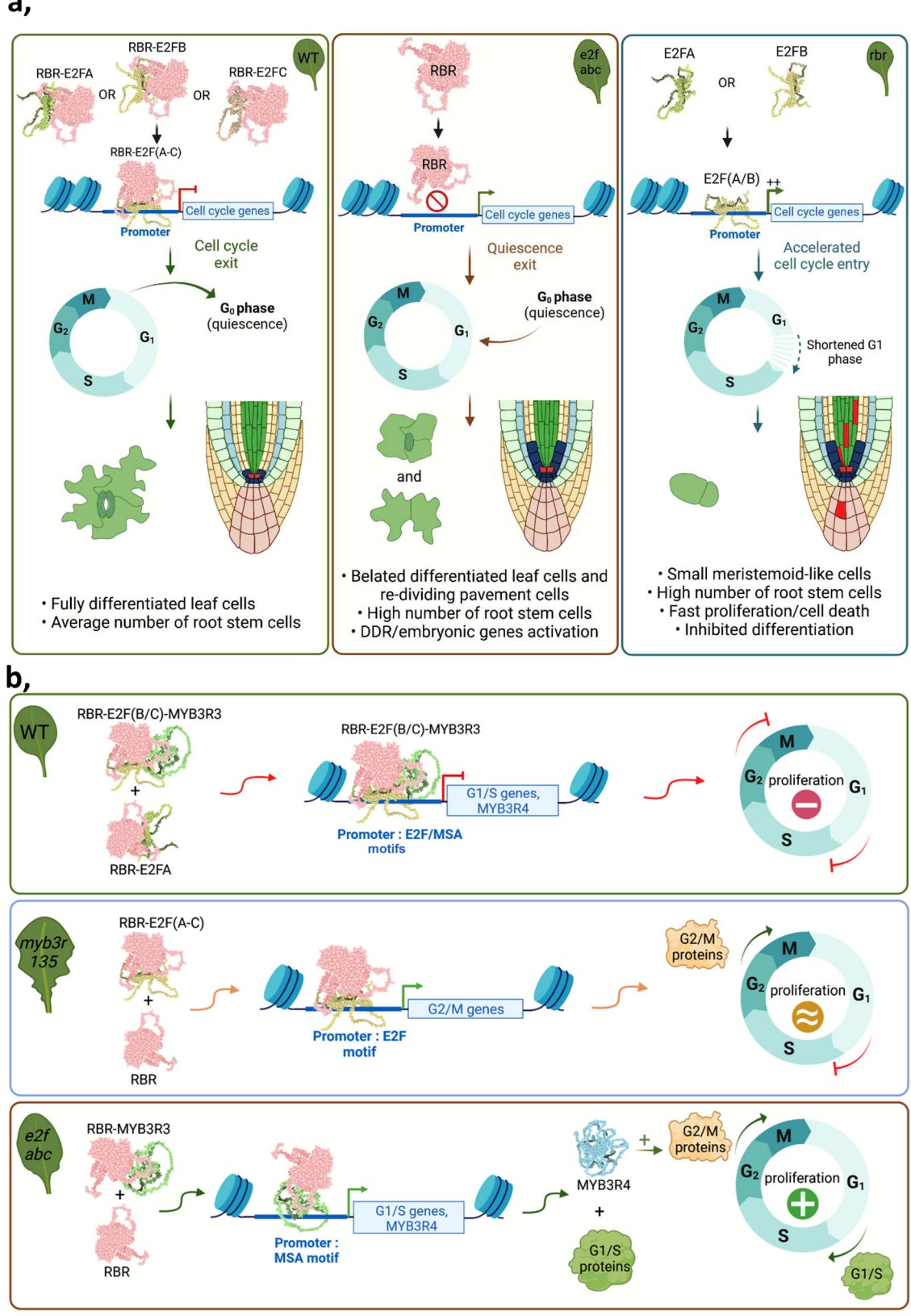

and RBR were specifically enriched on the promoter of the activator MYB3R4, a key regulator of mitosis, and indeed it was activated in the *e2fabc* mutant leaf, which might explain the central role of E2Fs to maintain cellular quiescence by controlling both G1/S and G2/M transitions (Fig. 6b). Recently a similar feed forward regulatory loop has been discovered in the mammalian cell cycle, where the activity of G1/S regulator CDK4/6 was also responsible to maintain CDK2 throughout G2/M and thus the cell cycle re-activation and restriction point remains sensitive to mitogenic signals until mitosis[55].

Taken together, we have shown that the canonical E2Fs together with RBR form a repressor complex on a cohort of cell cycle

**Fig. 6 Schematic model how E2Fs and RBR induce developmental quiescence. a** Canonical E2Fs function as co-repressors of cell cycle genes with RBR. Left: in the wild-type, E2Fs form complexes with RBR to repress cell cycle genes and allow cell division arrest in quiescent and differentiated cells. Middle: in the *e2fabc* mutant, lack of RBR-E2F complexes results in de-repression of cell cycle genes leading to a delay in cell differentiation as well as division reactivation in quiescent cells. Right: in RBR-deficient lines, E2F/RBR complexes are also lacking, but E2Fs are fully active, leading to a more pronounced hyperactivation of cell cycle genes and thus completely preventing cell differentiation, and inducing cell death. **b** Distinct molecular roles of E2Fs and MYB3Rs in the control of cell cycle arrest. Top: in the wild-type, Repressor complexes containing RBR, E2Fs and MYB3Rs target G1/S and G2/M genes to repress their expression either via the MSA or the E2F motifs. Middle: in the *myb3r1,3,5* mutant that lacks repressor MYB3Rs, E2Fs are still present and able to repress G1/S and some G2/M genes bound via the E2F motif. Cell proliferation is therefore not induced in differentiated cells. Bottom: in the *e2fabc* mutant, repressor MYB3Rs can still repress some G2/M genes with or without RBR, but repression of both G1/S and G2/M activators such as MYB3R4 is lost, resulting in enhanced cell proliferation. Model was created by using Biorender.

genes that is pivotal for maintaining quiescence. Through this repression mechanism it is possible to influence plant growth, which could open novel ways for improving crop biomass and productivity.

## Methods

**Plant material and growth conditions**. Arabidopsis thaliana Col-0 ecotype was the WT and background of all transgenic lines used in this study. In vitro-cultured plants were grown on a half-strength germination medium under continuous light at 22 °C. Soil grown plants were cultivated in a greenhouse at 22 °C under long-day conditions (16 h light/8 h dark). Most of the transgenic and T-DNA insertion mutant lines used in this study have been previously published: E2FA-GFP, RBR-GFP[16], E2FB-GFP[17], E2FC-GFP[36], CYCB1;2-YFP[56], *e2fb-1* (SALK_103138), *e2fa-2* (GABI-348E09), *e2fc-1*, (GK-718E12,[20,48]; the double *e2fab* (*e2fa-2/e2fb-1*) was reported by Heyman et al.[19], and the triple *e2fabc* (*e2fa-2/e2fb-1/e2fc-1*) was generated by Wang et al.[20]. The second T-DNA insertion line for E2FC, named as *e2fc-2* (SAIL-1216G10) was obtained from the SALK Institute and the *e2fabc-2* triple mutant was generated by crossing the *e2fab* double (*e2fa-2/e2fb-1*) with the *e2fc-2*. By crossing the corresponding E2F mutants we also generated *e2fc-1/e2fa-2* and *e2fc-1/e2fb-1* double mutants as well as we also generated an *e2fabc* triple mutant line expressing the CYCB1;2-YFP marker. RBR co-suppression (*RBRcs*) seedlings were identified in transgenic lines expressing either the RBR-3xCFP[18] or the RBR-RFP[57] in the WT background under the control of the native RBR promoter showing strong growth arrested phenotype identical with previously reported[9].

Young seedlings (5, 7, or 14 days after germination, DAG) or the first true leaf pairs of the wild-type and mutant Arabidopsis lines (*e2fabc* and *e2fab*) grown in vitro were harvested 9, 12, 16 and 20 DAG, flash frozen and stored at −80 °C.

**Microscopy, EdU labelling and flow cytometry analysis**. Mature dried seeds were imbibed for 1 h and dissected under the stereomicroscope. Isolated embryos were stained with propidium iodide (PI) and photographed under confocal laser microscopy (SP5, Leica). Organ and epidermal cell sizes were measured using ImageJ software[18,58]. Cotyledons and the first true leaf pairs of wild-type and mutant lines were dissected from seedlings at 4–8 DAG and 8-16 DAG, respectively. Leaves were stained with propidium iodide (PI; 20 μg/mL) and images on the abaxial side of the cotyledon or the leaf were taken and analysed by confocal laser microscopy. 600–300 cells were counted and measured per cotyledon or leaf samples (*n* = 3 biological replicates and *N* = 5 samples in each case were studied for the transgenic lines and the control WT) using ImageJ software. Average cell size was calculated, and the total cell number was extrapolated to the whole cotyledon and the first real leaf pair[17,59]. To visualise the distributions of the cell area, only non-guard epidermal cells were used for calculation[59]. The number

of elongated pavement cells with newly formed cell wall (described as extra cell division) was counted in three different zones (the basal, middle and tip parts) and extrapolated to the whole leaf. Roots were also visualised after PI staining (20 μg/mL) under confocal laser microscopy[17].

For DIC microscopy of cleared tissues, plant tissues were fixed in a 9:1 of ethanol and acetic acid solution and cleared with Hoyer's solution (a mixture of 100 g chloral hydrate, 10 g glycerol, 15 g gum arabic, and 25 mL water). They were mounted onto glass slides and used for microscopic observations with differential interference contrast (DIC) microscope (BX51, Olympus). Images were captured with a CCD camera (DP74, Olympus) and an image capture software (CellSens Standard, Olympus). For visualisation of starch granules in columella cells, roots were incubated for 3 min in Lugol solution (Sigma) before clearing.

For EdU incorporation assay WT and the mutant seedlings were grown in half strength liquid MS containing 10 μM EdU (Click-iT Alexa Fluor 647 Imaging Kit; Invitrogene) and incubated either for an hour or for 24 h. Afterward, seedlings were treated according to[60], and the root samples were also stained by DAPI solution and the observations were done under the confocal laser microscope.

For flow cytometry measurement, the first leaf pair and the cotyledons were collected and chopped by razor blades in nuclei extraction buffer and stained with DAPI with the CyStain UV Precise Kit (Partec, Magyar et al., 2005). Nuclear DNA content was determined by using Partec PAS2 Particle Analysing system (Partec, Germany).

**cDNA preparation and RT-qPCR**. RNA was extracted from the plant material using a CTAB-LiCl method of[61]. RNA samples were treated with DNase1 (ThermoScientific #EN0521) according to the manufacturer's protocol. cDNA was synthesised using 1 μg of RNA using the ThermoScientific Reverse Transcription Kit (#K1691) with random hexamers based on the manufacturer's prescription. Mock reaction without RevertAid enzyme was also prepared to ensure that there is no contaminating genomic DNA in the samples. RT-qPCR in the presence of SYBR Green (TaKaRa TB Green Primer Ex TaqII, #RR820Q) was carried out according to the manufacturer's instructions in a BioRAD CFX 384 Thermal Cycler (BioRAD) with the following setup: 50 °C 2 min, 95 °C 10 min, 95 °C 15 s, 60 °C 1 min, 40 cycles followed by melting point analysis. Each reaction was carried out in tree technical replicates and reaction specificity was confirmed by the presence of a single peak in the melting curve. All the data were normalised to the average Ct value of two housekeeping genes (ACTIN and UBIQUITIN) unless otherwise mentioned and the calculated efficiency was added to the analysis. Amplification efficiencies were derived from the slope of amplification curves at the exponential phase. Primer sequences are summarised in Supplementary Table 1.

**Immunoprecipitation and immunoblotting**. Total proteins were extracted from dissected leaves or young seedlings in extraction buffer (25 mM Tris-HCl, pH 7.5, 75 mM NaCl, 15 mM MgCl2, 15 mM EGTA, 15 mM p-nitrophenylphosphate, 60 mM β-glycer-ophosphate, 1 mM dithiothreitol, 0.1% (v/v) IGEPAL CA-630, 0.5 mM NaF, 1 mM phenylmethylsulfonyl fluoride, and protease inhibitor cocktail for plant tissue [P9599, Sigma]). For immuno-precipitation (IP), equal amounts of protein samples (800 µg) in the extraction buffer[17] were incubated with anti-DPB antibodies for 1 h at 4 °C on a rotary shaker. Dynabeads Protein A (Invitrogen) was used to pull down polyclonal antibodies, and after washing the beads proteins were eluted by adding SDS sample buffer followed by 5 min boiling. Eluted proteins were loaded on SDS-PAGE gels (10%) and after protein gel electrophoresis proteins were trans-ferred to polyvinylidene difluoride membrane (PVDF, Milipore) and afterwards immunoblotting assay was carried out. For immu-noblotting assay[17], equal amounts of proteins were loaded to SDS-PAGE gel (10%), and proteins were transferred onto PVDF membranes. The membranes were blocked in 5% (w/v) milk powder with 0.05% (v/v) Tween 20 in Tris-buffered saline (TBS; 25 mM Tris-Cl, pH 8.0, and 150 mM NaCl; TBS plus Tween 20 [TBST]) buffer for 1 h at room temperature. The membrane was incubated with 5% (w/v) milk-powder TBST containing the pri-mary antibodies and agitated overnight at 4 °C.

Primary antibodies used in immunoblotting experiments were chicken anti-RBR antibody (1:2000 dilution; Agrisera), mouse monoclonal anti-PSTAIRE (1:40,000 dilution, CDKA;1 specific; Sigma), anti-phospho-specific Rb (Ser-807/811) rabbit polyclonal antibody (1:500 dilution; Cell Signalling Tech), anti-DPA, anti-E2FB polyclonal rabbit antibodies (both 1:400 dilution, Magyar et al., 2005)[45], anti-DPB polyclonal rabbit antibody[62], and anti-E2FA rat polyclonal antibody (1:400 dilution,[18], anti-12S globulin rabbit polyclonal antibody (1:10,000;[63]. After the primary antibody reaction, the membrane was washed (TBST) and incubated with the appropriate secondary antibody conjugated with horseradish peroxidase at room temperature. Afterward, chemiluminescence substrate was applied according to the manufacturer description (SuperSignal West Pico Plus, Thermo Fisher Scientific) or Immobilon western horseradish peroxidase (Millipore).

**Chromatin immunoprecipitation followed by high-throughput sequencing (ChIP-seq) assay**. ChIP-seq assays were performed on 14-d-old seedlings using anti-GFP antibody (Abcam). Plant material was cross-linked in 1% (v/v) formaldehyde for 15 min under vacuum at room temperature and cross-linking was then quenched with 0.125 M glycine for 5 min. Cross-linked plantlets were ground in liquid nitrogen, and nuclei were lysed in Nuclei Lysis Buffer (0.1% SDS, 50 mm Tris-HCl at pH 8, 10 mm EDTA, pH 8). Chromatin was then sonicated for 5 min using a Covaris S220 (Peak Power: 175, cycles/burst: 200. Duty Factory: 20). Immunoprecipitation was performed overnight at 4 °C with gentle shaking, and immuno-complexes were next incubated for 1 h at 4 °C with 40 µL of Dynabeads Protein A (Thermo Fisher Scientific). The beads were washed 2 × 5 min in ChIP Wash Buffer 1 (0.1% SDS, 1% Triton X-100, 20 mM Tris−HCl at pH 8, 2 mM EDTA at pH 8, 150 mM NaCl), 2 × 5 min in ChIP Wash Buffer 2 (0.1% SDS, 1% Triton X-100, 20 mM Tris−HCl at pH 8, 2 mM EDTA at pH 8, 500 mM NaCl), 2 × 5 min in ChIP Wash Buffer 3 (0.25 mM LiCl, 1% NP-40, 1% sodium deoxycholate, 10 mM Tris−HCl at pH 8, 1 mM EDTA at pH 8) and twice in TE (10 mm Tris-HCl at pH 8, 1 mM EDTA at pH 8). ChIPed DNA was eluted by two 15-min incubations at 65 °C with 250 µL of Elution Buffer (1% SDS, 0.1 m NaHCO₃). Chromatin was reverse cross-linked by adding 20 µL of NaCl 5 M and incubated over-night at 65 °C. Reverse cross-linked DNA was treated with RNase and Proteinase K and extracted with phenol−chloroform. DNA was ethanol precipitated in the presence of 20 µg of glycogen and resuspended in 10 µL of nuclease-free water in a DNA low-bind tube. Libraries were then generated using 10 ng of DNA and NEBNext Ultra II DNA Library Prep Kit for Illumina (NEB), following the manufacturer's instructions. The quality of libraries was assessed with an Agilent 2100 Bioanalyzer (Agilent), prior to 1 × 75 bp high-throughput sequencing by NextSeq 500 (Illumina).

**Analysis of ChIP-seq data**. Trimmomatic-0.38 was used for quality trimming[64]. Parameters for read quality filtering were set as follows: minimum length of 36 bp; mean Phred quality score greater than 30; leading and trailing bases removal with base quality <5. The reads were mapped onto the TAIR10 assembly using Bowtie 2[65] with mismatch permission of 1 bp. To identify significantly enriched regions, we used MACS2[66]. Parameters for peaks detection were set as follows: number of duplicate reads at a location:1; mfold of 5:50; Q-value cutoff: 0.05; extsize 200; sharp peak. Average scores across genomic regions were extracted using the multiBigwigSummary command of the DeepTools package after data normalisation with the S3norm software[67]. Data were visualised using WashU and the plot Heatmap tool of the DeepTools package[68] was used to generate heatmaps while the ggplot2 package was used to draw metaplots.

**Motif enrichment analysis**. Position weight matrix (PWM) data for binding motifs of E2FA and MYB3R5, determined by DNA affinity purification sequencing (DAP-Seq), were obtained from a website of Plant Cistrome Database at http://neomorph.salk.edu/PlantCistromeDB[69], and used as E2F- and MYB3R-binding motifs. Motif search based on PWM was conducted using the matchPWM function implementing in R Biostrings package. Forward and reverse of each motif were used as PWM with minimum score of 50%.

**Transcriptome analysis**. Total RNAs were extracted using the RNeasy Plant Mini Kit (QIAGEN, Germany) from whole seed-lings according to the manufacturer's instructions. Total RNAs from WT and e2fabc at 9 DAG were used for construction of cDNA libraries using the TruSeq RNA Library Preparation Kit v2 (Illumina, United States) according to the manufacturer's pro-tocol. For transcriptome profiling in WT and e2fabc plants, we analysed three biological replicates for statistical analysis. The libraries were sequenced using the NextSeq500 sequencer (Illu-mina, United States). Raw reads containing adaptor sequences were trimmed using bcl2fastq (Illumina, United States), and nucleotides with low-quality (QV < 25) were masked by N using the original script. Reads shorter than 50 bp were discarded, and the remaining reads were mapped to the cDNA reference using Bowtie with the following parameters: "–all–best–strata"[70]. The reads were counted by transcript models. Differentially expressed genes were selected based on the adjusted P-value calculated using edgeR (version 3.20.9) with default settings[71].

**Other methods**. Gene ontology analysis was carried out using the singular enrichment analysis tool offered by agriGO with the default settings[72].

**Statistics and reproducibility**. Each experiments were performed with three biologically independent samples unless stated other-wise. Statistical analysis between two groups was analysed by a paired t test in Microsoft Excel. The p values were all two-tailed, and $p < 0.05$ was considered significant: *$p < 0.05$, **$p < 0.01$, ***$p < 0.001$. All experiments in this study were repeated three times.

## Reporting summary

**Reporting summary**. Further information on research design is available in the Nature Portfolio Reporting Summary linked to this article.

## Data availability

RNA-Seq data for WT and *e2fabc* can be accessed from the DDBJ database under accession number DRA016858. ChIP-Seq data of RBR, E2FA, E2FB and E2FC can be accessed at Gene Expression Omnibus database under accession number GSE218481. The source data are provided with this paper including raw western blots (Supplementary Data 4 and Supplementary Fig. 14). Other source data supporting the findings of this study and Arabidopsis mutants and transgenic lines generated and used in this study are available from the corresponding author upon reasonable request.

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

## Acknowledgements
We thank Xinnian Dong for the *e2fabc* mutant line, Arp Schnittger for the RBR-RFP line. This work was supported by a grant from the Hungarian National Research Funding (NKFI-139202) to Z. Magyar, the Japan Society for the Promotion of Science KAKENHI (22K06261 and 22H04714) to M. Ito, Japan Science and Technology Agency (JST grant number JPMJPF2102) to M. Ito, Biotechnology and Biological Sciences Research Council BBSRC-NSF grant BB/M025047/1 to L. Bogre and Cs Papdi. We are grateful for Attila Fehér for the critical reading and helpful discussion.

## Author contributions
Z.M., L.B., M.I., C.R. and M.B. conceived the study; M.G., C.R, Z.M., M.I. designed the experiments; M.G., C.R., Z.M., Y.N., E.M., R.B.-C., H.T., A.Z., D.B., D.L., K.M., F.N., H.I., E.Ő. and C.P. carried out the experiments; X.H., J.A. and T.S. analysed the data, C.B. did the model; Z.M., C.R., M.G., L.B., M.I. and M.B. wrote the paper.

## Funding

## Competing interests
The authors declare no competing interests.
