## [Peer Review File · Communications Biology]

Reviewers' comments:

Reviewer #1 (Remarks to the Author):

I appreciate the efforts made by the authors to address my comments on their work aimed at defining a role of Arabidopsis E2FA, B and C in restricting cell proliferation after the switch to quiescence. The study is largely based on the phenotypic analysis of a e2fabc triple mutant that shows overproliferation not only by increasing the number of dividing cells (root meristems) but also by the formation of extra cell walls in endoreduplicated pavement cells (leaves). In the revised version authors elaborate on the major claim of their study, that canonical E2F factors in complex with RBR establish cellular quiescence by repressing cell cycle gene expression. There are a number of points that still remain obscure or not sufficiently justified in the authors rebuttal.

1. Cell division results in different organs. I fully recognize the value of looking at various organs. However, some phenotypes show discrepancies among them. For example, the extra columella stem cell layers is not observed in other cell layers. Also, the ectopic division of differentiated pavement cells is not observed in the differentiated root cells. In spite that high RBR-P levels are detected in old leaves, those discrepancies need a stronger support and/or explanation.

2. Ext. Data Fig 5. The formal conclusion of these results must be that E2FC plays a central role in restricting leaf area and CYCB1;1 expression. Is this the case? Any claim on redundancy would probably need other experiments.

3. Ext. Data Fig. 4b, also Ext. Data Fig. 10. The pattern of CYCB1;1-YFP is awkward. Please justify the appearance of this cyclin in the two nuclei. Also, this pavement cell is relatively small compared with others, suggesting that is likely 4C, perhaps 8C. Are the new cells the result of a reduction in DNA content or do they actually undergo a normal S-phase? The appearance of extra cell walls (extra divisions) was also observed in plants overexpressing E2Fa and DPA. How do you reconcile these observations with a similar phenotype in e2fabc triple mutant? All these issues need to be addressed directly and described carefully in the text.

4. Ext. Data Fig. 4d. The endocycle appears to be stimulated in leaves but repressed in cotyledons. Please, explain this discrepancy.

5. The issue regarding the direct link proposed between loss of the three E2Fs and the phenotype unrestricted quiescence is not sufficiently addressed. As indicated in my previous report. Many of the genes upregulated are not only "cell cycle" but also "endocycle" genes. The effects might also be dependent on the (many) other genes misregulated in the triple mutant. These possibilities are not even discussed in detail. Therefore, the claims must be adapted to consider alternative explanations.

Regarding the comments of reviewer #2:

Point 1. The differences observed between the double e2fab and the triple e2fabc mutants still leave many open questions that have not been addressed by the authors.

Point 2. A detailed analysis and, ideally extra experiments, would be necessary to fully support the authors' claim that all the phenotypic effects are due to the lack of the three E2F directly and not to the may non-E2F target genes misregulated in the triple mutant.

Reviewer #3 (Remarks to the Author):

Within this work it is demonstrated that the three Arabidopsis E2F proteins together are required for developmental controlled cell cycle exit, likely through their recruitment of the RBR protein. It is a reworked version of a previous submission in which the authors have corrected the many wrong or missing references to figures, and significantly improved the figures and their legends. There is little addition of new data compared to the original version, but that version was already stuffed with data. The problem was rather that these were poorly presented and described. With this problem being solved, I believe that the findings are of sufficient novelty and interest to be published.

Remaining remarks:

- I still believe it is impossible to say that the epidermal cells re-entered the cell cycle. Rather the cells never lost their division potential, even despite showing the typical lobbed shape of pavement cells. Accordingly, the novel ploidy figure (Extended data Fig. 4d) shows that barely any cell with a DNA content above 4C can be detected. With endoreduplication being a marker of cell differentiation, in case the authors want to cling to the re-entry hypothesis they may try showing that the cells displaying CYCB1;2 expression have a DNA content being at least 4C (thus being the product of a divided 8C cell).
- Although the authors may retain their hypothesis on the difference between the *rbr* and *e2fab* mutants, it cannot be ignored that in the latter there is still some recruitment of RBR (Fig 4a), likely due to the truncated E2Fa still having some RBR binding potential. This recruitment might be sufficient to cause the difference between the two genotypes. At the possibility of this alternative hypothesis needs to be mentioned.

Reviewer #1:

I appreciate the efforts made by the authors to address my comments on their work aimed at defining a role of Arabidopsis E2FA, B and C in restricting cell proliferation after the switch to quiescence. The study is largely based on the phenotypic analysis of a e2fabc triple mutant that shows overproliferation not only by increasing the number of dividing cells (root meristems) but also by the formation of extra cell walls in endoreduplicated pavement cells (leaves). In the revised version authors elaborate on the major claim of their study, that canonical E2F factors in complex with RBR establish cellular quiescence by repressing cell cycle gene expression. There are a number of points that still remain obscure or not sufficiently justified in the authors rebuttal.

1. Cell division results in different organs. I fully recognize the value of looking at various organs. However, some phenotypes show discrepancies among them. For example, the extra columella stem cell layers is not observed in other cell layers. Also, the ectopic division of differentiated pavement cells is not observed in the differentiated root cells. In spite that high RBR-P levels are detected in old leaves, those discrepancies need a stronger support and/or explanation.

We agree that quiescence was not disrupted in each of the studied organs and cell types exactly the same way in the e2fabc mutant, but we do not consider this as a discrepancy, rather reflecting that different cell types have various levels of quiescence dependent on the developmental context, developmental regulation, positional organisation of stem cells and cell files. The emerging model in animals is that the Rb-E2F constitutes a bistable switch with two stable states; E2F-On and OFF, stimulating or repressing the cell cycle, respectively. Interestingly, the serum amount/strength required to turn On the switch varies from one cell to another, indicating that a unique E2F activation threshold defines each quiescent state (Yao et al., 2014).

Terminally differentiated cells in Drosophila wings and eyes for example are largely resistant to proliferation upon deregulation of either dE2F or Cyclin E, but exogenous expression of both factors together can bypass cell cycle exit (Buttitta et al., 2010).

It is known that leaf has determinate, while root has indeterminate organ development, and consequently they have different temporal and spatial cell

proliferation patterns under the control of the E2F-RBR (Wildwater et al., 2005; Ószi et al., 2020). Therefore, we suggest that E2F-RBR might be involved to establish and/or maintain quiescence in both differentiated leaf and root cells but the requirement to stimulate exit from quiescence could be different, reflecting that the depth of quiescence might vary in the leaf and in the root. Further study is required to understand the molecular regulation of this process.

We showed that in the root stem cell niche region, the quiescent centre cells, and all the neighbouring stem cell initials including the columella stem cells became more proliferative in the triple *e2fabc* mutant in comparison to the WT (Fig.4e,g; Ext. Data Fig11., 12). Since leaf stomatal meristemoid cells proliferate more frequently in the triple *e2fabc* mutant leaf we suggest that the transient quiescence cellular status is under the control of the E2F-RBR repressors.

We have added these additional considerations above in the discussion part of the manuscript.

2. Ext. Data Fig 5. The formal conclusion of these results must be that E2FC plays a central role in restricting leaf area and CYCB1;1 expression. Is this the case? Any claim on redundancy would probably need other experiments.

We showed that neither *e2fc-1* single mutant nor its double mutant combinations with *e2fa-2* (*e2fac-1/e2fa-2*) or *e2fb-1* (*e2fc-1/e2fb-1*) resulted in overproliferation in the embryo in respect to WT control. In these single and double *e2fc* mutant lines we also did not observe the characteristic extra divisions in the hypocotyl epidermis, but only in the *e2fabc* triple mutant (Ext. Data Fig.3f). Based on this we concluded that E2FC together with the two other canonical E2Fs redundantly control quiescence in complex with RBR.

3. Ext. Data Fig. 4b, also Ext. Data Fig. 10. The pattern of CYCB1;1-YFP is awkward. Please justify the appearance of this cyclin in the two nuclei. Also, this pavement cell is relatively small compared with others, suggesting that is likely 4C, perhaps 8C. Are the new cells the result of a reduction in DNA content or do they actually undergo a normal S-phase? The appearance of extra cell walls (extra divisions) was also observed in plants overexpressing E2Fa and DPA. How do you reconcile these observations with a similar phenotype in *e2fabc* triple mutant? All these issues need to be addressed directly and described carefully in the text.

We agree with the reviewer as we explained in the previous response that the CYCB1;2-YFP spuriously remains present till telophase in the cells, does not reflect the natural situation for CYCB1;2 endogenous protein at this point of the cell cycle, but it is nevertheless informative as a marker for newly divided cells. We added a comment on this to the manuscript. The ectopic co-expression of E2FA and DPA as well as ectopic E2FB under the control of its own promoter could stimulate the division of differentiated pavement cells (De Veylder et al., 2002; Ószi et al., 2020) supporting that they could escape from RBR repression. In the triple *e2fab*c mutant, all E2Fs lost their transactivation function therefore this function is not required for triggering cell proliferation in differentiated cells, rather the escape from RBR repression that matters.

4. Ext. Data Fig. 4d. The endocycle appears to be stimulated in leaves but repressed in cotyledons. Please, explain this discrepancy.

We do not think that endocycle was upregulated either in the cotyledon or in the leaf. In both cases there are a clear and significant enrichment of 4C cells, and their cellular phenotypes are very similar to each other, indicating overproliferation (Fig. 2 and 4). To make the point more clear that there is a G2 arrest both in leaf and cotyledon of the *e2fab*c triple mutant, we decided to show an earlier time point of 14d, when in WT the entry into endocycle just started (Extended data Fig. 4d).

5. The issue regarding the direct link proposed between loss of the three E2Fs and the phenotype un restricted quiescence is not sufficiently addressed. As indicated in my previous report. Many of the genes upregulated are not only “cell cycle” but also “endocycle” genes. The effects might also be dependent on the (many) other genes misregulated in the triple mutant. These possibilities are not even discussed in detail. Therefore, the claims must be adapted to consider alternative explanations.

Most of the S-phase genes supposed to be activated both in endoreduplicating and mitotic cells, therefore on the basis of gene expression it is really difficult to decide whether mitosis or endocycle was activated. Expression of activators of APC/C like the CCS52A1 involved in the switch from mitosis to endocycle did not change in the triple *e2fab*c mutant in comparison to the WT (Figure 1). On the other hand, endocycle is characteristic for differentiated cells, and elevated ploidy

level usually increases cell size of terminally differentiated cells, and did not increase cell number. We agree with reviewer that S-phase genes are misregulated in the triple mutant but we do not think that endocycle was up-regulated since cell size was not increased, while cell number was elevated.

Figure 1. Q-RT-PCR analyses of the expression of *CCS52A1* in developing first leaf pairs of *e2fab* and *e2fabc* mutants in comparison to the WT control.

In regard to Reviewer #2:

Point 1. The differences observed between the double *e2fab* and the triple *e2fabc* mutants still leave many open questions that have not been addressed by the authors.

Overproliferation is exclusively observed in the triple *e2fabc*, and neither the single nor the double *e2f* mutants showed similar overproliferation phenotype. Therefore we think that E2Fs redundantly required for repression of cell proliferation and imposing quiescence.

Point 2. A detailed analysis and, ideally extra experiments, would be necessary to fully support the authors' claim that all the phenotypic effects are due to the lack of the three E2F directly and not to the may non-E2F target genes misregulated in the triple mutant.

As we explained in our previous response to reviewer 2, it is not surprising to observe that many misregulated genes are not direct target. However, these

indirectly deregulated genes are also linked to the *e2fab* mutation. It is not clear to us what type of experiments one would ideally need to do to dissect the contribution of direct and indirect cohorts of genes to the overproliferation phenotypes in the *e2fab* triple mutant, but it would certainly be outside the scope of the present work.

Reviewer #3 (Remarks to the Author):

Within this work it is demonstrated that the three Arabidopsis E2F proteins together are required for developmental controlled cell cycle exit, likely through their recruitment of the RBR protein. It is a reworked version of a previous submission in which the authors have corrected the many wrong or missing references to figures, and significantly improved the figures and their legends. There is little addition of new data compared to the original version, but that version was already stuffed with data. The problem was rather that these were poorly presented and described. With this problem being solved, I believe that the findings are of sufficient novelty and interest to be published.

Remaining remarks:

- I still believe it is impossible to say that the epidermal cells re-entered the cell cycle. Rather the cells never lost their division potential, even despite showing the typical lobbed shape of pavement cells. Accordingly, the novel ploidy figure (Extended data Fig. 4d) shows that barely any cell with a DNA content above 4C can be detected. With endoreduplication being a marker of cell differentiation, in case the authors want to cling to the re-entry hypothesis they may try showing that the cells displaying *CYCB1;2* expression have a DNA content being at least 4C (thus being the product of a divided 8C cell).

There is precedence that e.g. differentiated root hair cells, having gone through endocycle, can re-enter cell division <https://www.nature.com/articles/nplants201589> . However, we do not know examples where proliferating cells would enter into cellular differentiation, quite the opposite, increased proliferation typically coupled with inhibited differentiation. A clear example is the *rbr* mutant where there is high overproliferation and inhibition to exit into differentiation. We agree with this

reviewer that it is difficult to experimentally determine whether differentiated leaf cells regain or retain their proliferation activity. Therefore, we added these points to the discussion.

- Although the authors may retain their hypothesis on the difference between the *rbr* and *e2fab*c mutants, it cannot be ignored that in the latter there is still some recruitment of RBR (Fig 4a), likely due to the truncated E2Fa still having some RBR binding potential. This recruitment might be sufficient to cause the difference between the two genotypes. At the possibility of this alternative hypothesis needs to be mentioned.

We agree with reviewer that the truncated E2FA mutant protein might have RBR binding activity since it has an intact MARKED BOX (MB) region, which was shown in animals to be able to interact with Rb protein. Previously, we overexpressed GFP-tagged mutant E2FA or E2FB with intact MB region and we were unable to precipitate RBR protein (Ószi et al., 2020). However, by using anti-DPB antibody we could still precipitate a very small amount of RBR protein from the triple *e2fab*c mutant, indicating that it still has some RBR-binding capacity. We added this point to our discussion.

REVIEWERS' COMMENTS:

Reviewer #1 (Remarks to the Author):

As in the previous revision, I appreciate the efforts to address my points. While this is true, I also maintain that some concerns (as also pointed out by another reviewer) there are some weak points. This is mostly due to the use of terms that are heavily misleading, e.g., re-entry cell division or overproliferation (see below).

Point 3. What is the fraction of cells that re-enter the cell cycle (using the authors' terminology), identified by the presence of a new cell wall, that continue dividing? In other words, how many of the new daughter cells develop a full cell division cycle, and form a new cell wall? It seems that this fraction is low. In that case, I wonder if it is fully correct to make a general statement such that they re-enter the cell cycle if the vast majority develop a cytokinesis just once. I suggest to take this into account in the text.

Point 4. The term overproliferation is also misleading in the sense that a relatively small amount of cells actually develop repeated cell division cycles, which would lead to a considerable increase in cell number. As indicated above, authors should take this into account reducing the strength of the terms used and the subsequent claims.

Reviewer #3 (Remarks to the Author):

I'm fine with the revisions. However, I still doubt on the newly added statement between lines 343-346 that refers to work illustrating a lack of binding between RBR and truncated E2Fs. However, Fig. 4a still shows a clear RBR precipitation band in the e2fabc mutant background. Hence, an interaction between the truncated E2FA and RBR is very likely.

Reviewer #1 (Remarks to the Author):

As in the previous revision, I appreciate the efforts to address my points. While this is true, I also maintain that some concerns (as also pointed out by another reviewer) there are some weak points. This is mostly due to the use of terms that are heavily misleading, e.g., re-entry cell division or overproliferation (see below).

Point 3. What is the fraction of cells that re-enter the cell cycle (using the authors' terminology), identified by the presence of a new cell wall, that continue dividing? In other words, how many of the new daughter cells develop a full cell division cycle, and form a new cell wall? It seems that this fraction is low. In that case, I wonder if it is fully correct to make a general statement such that they re-enter the cell cycle if the vast majority develop a cytokinesis just once. I suggest to take this into account in the text.

In regard of terminology, we use the term of re-entry into cell division as defined by other authors in the animal field when quiescence is broken, differentiated cells re-enter the cell division cycle ^{1,2}. Overproliferation or excessive proliferation are used synonymously when it refers to deregulated cell cycle that result in supernumerary (but non-quantitated) numbers of cells. To satisfy the reviewer we stick to excessive proliferation throughout the manuscript. When we studied this in the context of organ development we call this hyperplasia meaning supernumerary cells that integrated in a larger organ. We have calculated the number of new cell divisions of pavement cells to the entire leaf epidermis in the triple *e2fabc* by counting and extrapolating the number of straight cell walls, and it was estimated to be over 4000 at 16DAG resulted in over 8000 extra more cells, which would increase cell number by over 40% in the WT at that leaf developmental stage (Fig 2F). We also quantitated the re-entry of pavement cells (straight cell walls) in the case of cotyledon and found 77+/- 7% of pavement cells show this while it is essentially zero in the WT (Extended data Fig 7B, lines 216-17). We put forward a model (Fig 6), which shows that in the *e2fabc* triple the re-entry to cell division requires the regulation of both G1/S and G2/M control points. We show this molecularly (deregulation of key cell cycle regulators at these transitions) and in cell cycle terms by the EdU label, CYCB-GFP and straight cross walls. Recently a similar connection between G1/S and G2/M control was established in animal cells ³.

Point 4. The term overproliferation is also misleading in the sense that a relatively

small amount of cells actually develop repeated cell division cycles, which would lead to a considerable increase in cell number. As indicated above, authors should take this into account reducing the strength of the terms used and the subsequent claims.

The embryo, leaf and cotyledon of the triple *e2fab* mutant consist of much more cells than the WT control, and more root meristem cells entered into S-phase as well. In addition, quiescent stem cells including root QC cells divided more frequently in the triple *e2fab* than in the WT. All together these data strongly support that cells are more proliferative in the triple *e2fab* mutant in comparison to the control WT suggesting overproliferation of cells when they are normally quiescent in the WT. We show that the stomata meristemoid stem cell lineage do go through repeated cell division cycles to a much larger extent than in WT. Differentiated pavement cells re-enter cell division typically once, but we do detect in some large pavement cells multiple straight cross walls, meaning multiple divisions (Supplementary Fig. 4A, lines 144-46).

Reviewer #3 (Remarks to the Author):

I'm fine with the revisions. However, I still doubt on the newly added statement between lines 343-346 that refers to work illustrating a lack of binding between RBR and truncated E2Fs. However, Fig. 4a still shows a clear RBR precipitation band in the *e2fab* mutant background. Hence, an interaction between the truncated E2FA and RBR is very likely.

To satisfy the reviewer we made the following changes in the manuscript

Lines 210-11: The residual RBR binding in the *e2fab* mutant might be mediated by the MB region retained on E2FA in the case of *e2fa-2* mutant allele.

Lines 347-53: Because through the dimerization partner, DPB, we still could detect a tiny amount of RBR association in the triple *e2fab* mutant, presumably through

the conserved MB region that was retained on E2FA in the *e2fa-2* allele, we cannot rule out that a small residual E2F-RBR complex is able to repress cell proliferation to an extent that lead to cell cycle exit and differentiation. However, the RBR association in the *e2fab* is far less than in the double *e2fab* mutant, supporting that the main RBR binding activity of E2Fs is at the C-terminal RBR-binding domain, which in the case of *e2fab*, is presented on E2FC.

1. Marescal O, Cheeseman IM. Cellular Mechanisms and Regulation of Quiescence. *Dev Cell* **55**, 259-271 (2020).
2. Johnson MS, Cook JG. Cell cycle exits and U-turns: Quiescence as multiple reversible forms of arrest. *Fac Rev* **12**, 5 (2023).
3. Cornwell JA, Crncec A, Afifi MM, Tang K, Amin R, Cappell SD. Loss of CDK4/6 activity in S/G2 phase leads to cell cycle reversal. *Nature* **619**, 363-370 (2023).